# Self-Perception of Digital Competence in University Lecturers: A Comparative Study between Universities in Spain and Peru According to the DigCompEdu Model

Lorena Martín-Párraga *, Carmen Llorente-Cejudo and Julio Barroso-Osuna

Department of Didactics and Educational Organization (DOE), University of Seville, 41018 Seville, Spain; karen@us.es (C.L.-C.); jbarroso@us.es (J.B.-O.)
* Correspondence: lorena@grupotecnologiaeducativa.es

**Abstract:** Today's technology presents a major challenge for the education system in terms of digital literacy and the quality of education. This study focused on analyzing the level of digital competence of university teachers at the University of Seville, Spain, and another university in Arequipa, Peru, using the DigCompEdu model (Digital Competence Framework for Educators). The aim was to obtain significant data that reflect the educational reality within the use of Information and Communication Technologies (ICT) and to highlight the importance of training actions that improve the digital competence of teachers. The study was carried out using a descriptive–inferential approach, which showed the reality of the groups and obtained general conclusions based on the collected data. The participating teachers answered a self-assessment questionnaire that was previously validated by experts. The analyses and the obtained results highlight the importance of offering personalized training adapted to the needs of each educational context. Furthermore, they raise the need to reflect on areas for improvement and how to approach similar work in the future. It is suggested that future studies include probability samples in each research area to obtain more representative and globally applicable data.

**Keywords:** ICT; digital competence; competence frameworks; digital competence in teaching; DigCompEdu

## 1. Introduction

Social advances have generated a paradigm shift in terms of ICT, playing an essential role in educational innovation processes [1,2], which generates new training environments that guarantee an improvement in educational praxis, including quality pedagogical practices in these current transformation contexts, achieving an increase in social and educational digitalization [3]. Therefore, it will be crucial to take advantage of the opportunities provided by technological trends to carry out a transformation that turns the use of technology into an essential element to achieve innovative learning [4]. Focusing on the teaching and learning processes, several authors have studied the most relevant factors when it comes to offering a correct use of these by the teaching staff, promoting actions focused on the attitude shown by the teacher in the use and management of ICT [5,6], as well as actions aimed at their educational praxis [7], related to how a teacher assumes the possibility of integrating them into their teaching process as a factor that adds extra value to it [8]. In turn, this attitude is an essential conditioning factor in the processes of integrating ICT within teaching processes, as this can sometimes prevent progress as well as influence the motivation to participate in training activities. Different studies have shown that one of the conditioning factors is the interest of teachers in applying ICT in their daily lives [9].

The digital transformation, the modification of the teaching role, and the acquisition of digital competence (DC) are requests that are affecting all stages of the education system [10]. This diversity of changes has meant that educational institutions must modify

existing methodological plans, enabling their integration into educational practices [11,12]. Therefore, there is a need to offer a safe and critical use of ICT through DC training, which is considered one of the key competences that guarantee educational success. This is an ambiguous term with different connotations. Several studies agree on its complexity and diversity, and it includes key elements related to the management of technology, communication, and information processing [13].

This paper aimed to explore the educational reality and understand the importance of measuring and evaluating the level of DC of university teachers in different contexts, with the intention of identifying variables that influence this competence and proposing strategies for its future development, mainly focusing on the Spanish and Peruvian context. Different studies have supported the importance of analyzing the current situation of the digital competences of teachers to ensure a quality teaching process, taking into account educational problems in different educational contexts [14–16] and thus inculcating globalization, media, and information literacy [17–19].

### 1.1. The Reality of Education in Different Contexts

Different societies are immersed in a digital era that arouses unstoppable transformations. The implications generate greater technological mediatization, and that requires skills training to manage how to offer responses to these demands.

The different access gaps due to the deficient use of technology mean that rapid solutions are needed to provide the population with training responses, making this one of the focuses of the different political agendas and educational institutions at a global level [20]. In our context, there is a common line in educational policies, especially those focused on the development of digital competences in the educational sphere and on teachers as a starting point and a fundamental aspect in the empowerment of society and digital economic development.

According to a study carried out by the Organization for Economic Co-operation and Development (OECD) and the International Telecommunication Union (ITU) in 2018, these digital skills will be the cornerstone for the progress of societies in terms of training and skills as well as the possibility of ensuring that digital competences enable universal access to technology [21].

Focusing on the Spanish context, the educational reality is intrinsically related to this concern when it comes to establishing frameworks and actions in educational policies. Studies carried out by [21] show that in Spain, the formal education of teachers is more than double the education in other countries, such as in Latin America. This corresponds to the fact that the self-perception of Spanish teachers is very high, as detailed in the line of work of [22], who concluded that the higher the training received, the higher the self-perception. Despite this, if we make a comparison with countries with richer economies, according to the PISA 2018 data, Spain is among the countries where the use of ICT in education is least encouraged, at 53.3% compared to the OECD average of 67.2% [23]. This is why the reality differs from the self-perception. The Digital Spain 2025 Plan highlights the importance of establishing an ICT skills training program for teachers, the promotion of the Digital Teaching Competence (DTC) within the Framework of Reference for Digital Competence in Teaching, the promotion of actions in educational services, the creation of educational material, and the development of territorial cooperation policies through the National Institute of Educational Technologies and Teacher Training (INTEF) [24].

In the Peruvian context, the focus is on student competences, leaving aside teacher training. According to the Peruvian Ministry of Education (Minedu), the existence of a clear policy serving as a guideline for the use of technological resources that enable the digital restructuring of teachers is essential [25].

It was in 2017, with the implementation of the change in the basic education curriculum, that the importance of developing DTC was detailed, and with it, the need to acquire a framework of reference for the correct development of the CDD was identified. This led

to the development of policies to achieve these competences and, consequently, after their implementation, an improvement in the education system [26].

In 2020, the UN (United Nations) recommended supporting the preparation of these teachers to meet current challenges and to ensure equitable and inclusive learning in society.

In a study that analyzed the Peruvian virtual context, the need to strengthen actions using educational policies to achieve a strategic plan through a digital training program focused on the development of competency dimensions became [20], as the lack of existing competence among teachers, the resistance to incorporating new learning scenarios, the ineffectiveness of teacher training, the existing bureaucracy in organizations, and the lack of support from specialized staff for the empowerment of DTC [27] make it difficult to ensure the teacher's role as a digital manager and mediator.

Regardless of the context in which we find ourselves, the need to implement correct technological literacy as a tool to know how to introduce, manage, evaluate, and integrate technologies is notorious [28], making this an essential requirement when it comes to the correct searching and processing of information [29] as well as for the development of critical thinking focused on problem solving and decision making [30,31]. In the same way, there is a need to acquire skills that allow better development and the introduction of digital strategies oriented towards collaboration and the communication of information [32], which establishes a series of ethical notions that strengthen the use of good practices [33] and have an impact on a range of much more innovative and creative educational experiences [34].

In a study carried out by [35], the importance of considering the gender gap in educational contexts was detailed, as they affirmed the existence of a difference between the sexes based on perceived self-efficacy, which has been detailed in other studies [36]. In other words, there was a decrease in women's belief in their own abilities compared to men, with women perceiving themselves as less digitally competent [37–39]; therefore, greater interest is shown in improving their training. This is why it is important to foster, in these contexts, the belief in developing skills that favor the updating of these educational environments and thus be able to offer quality education in the use of technologies since, although there is progress, the simple fact of introducing ICT in teaching does not necessarily imply changes in learning environments [40].

As a result, the demands placed on education professionals to meet the wide range of changing challenges are becoming more and more pressing, requiring a wider range of competences [41]. The importance of acquiring digital training to ensure the acquisition of key digital competences to meet today's challenges is, therefore, understood.

Faced with this challenge, it is necessary to rethink not only the traditional educational model but also the places where learning takes place. This is why several initiatives have emerged to transform educational environments through the integration of technologies, changing the work approach, and reflecting on the educational use of digital competences [42].

*1.2. Frameworks and Instruments for the Evaluation of DTC*

The demands generated by the current transformation contexts add relevance to different key approaches, with the aim of achieving a correct improvement of the educational system in accordance with the current society's requests, which offers much more modern and extensible alternatives to this new social and educational reality in accordance with the recommendations exposed, at European level, about the need to introduce digital competences as an indispensable part of the educational system since these will achieve greater success.

To achieve European policy alignment, in 2017 the JRC presented the European Digital Competence Framework for Teachers (DigCompEdu), the product of research conducted at local, national, European, and international levels [43]. This framework was created with the main purpose of providing support to member states to promote DC and thus foster educational innovation.

DigCompEdu is a competency model made of six areas, which are associated with different competences that teachers need to acquire to foster productive and inclusive learning strategies through the use of digital tools [44].

Within this framework, we also find different levels of mastery. To be precise, there are a total of six levels of progressive mastery. The aim is to achieve the early detection of teaching competence levels and to enable gradual development and autonomy. It starts from an initial level (A1) and continues to a final level (C2).

The DigCompEdu Framework is also used as a reference in regional programs for the digital transformation of schools. At the same time, DigCompEdu is part of the Digital Teaching Competence Framework. This framework has been aligned with regional, national, and European proposals on digital competences, with the aim of incorporating the knowledge and experience acquired over the last few years and facilitating convergence in the creation of a European Education Area in 2025.

DigCompEdu Check-in

There are numerous studies working on how to adequately detect digital competence in teaching (DCD). In general, the most used instruments are questionnaires constructed in the form of Likert scales, which seek to collect the self-perceptions that teachers have about their own mastery of DCD. These instruments are usually created based on conceptual reviews carried out by the authors themselves [35] and are needed in competency frameworks formulated by institutions such as DigCompEdu [45].

DigCompEdu is a model that was created to develop a self-reflection tool for educators called "DigCompEdu Check-in" based on the European Framework of Digital Competence for Educators. The main objective of this questionnaire is to help educators improve their understanding of the European framework by providing them with a self-assessment of their strengths and weaknesses, which is crucial for educators to be highly competent in their professional practice.

The content of the DigCompEdu Check-In questionnaire has been incorporated into the EuSurvey self-assessment tool, which uses a global rating system organized by areas to identify the level of digital competence acquired by the educator.

The outcome of the model represents a consensus on the main areas and elements of DTC, following a progressive logic in each competence area. The main objective is to determine the level of self-perception shown by university teaching staff in both contexts and for this to serve to find significant differences between different groups of variables with respect to their level of competence, offering an identification of the degree of CDD. This is essential when it comes to incorporating training plans that guarantee an improvement in teaching practice and, with it, the quality of online teaching, as indicated by different studies on the subject [43,45].

## 2. Material and Methods

This study provides the basis for analyzing the level of digital competence of university teaching staff and inculcating the need to acquire key digital competences in line with existing demands.

This work is framed within ex post facto methodology research, which is carried out after an event has occurred, without altering the variables involved [46]. In fact, these are chosen according to the characteristics they possess. The researcher has no control over the variables. The effects have already occurred and are evidenced and/or recorded [47].

A descriptive–inferential design was proposed that included the participation of university professors from the two mentioned universities since in this type of study, data are collected to describe and characterize a particular phenomenon or population while, at the same time, generalizations are made about the wider population based on the results that are obtained [48]. We sought to collect detailed information on the studied groups through a Likert-type self-assessment questionnaire, which offered statistical validity and reliability, allowing us to obtain quantitative data on the teachers' responses. In addition,

statistical methods were used for analysis and interpretation beyond descriptive data. The aim was to obtain general conclusions and draw inferences about the reality of the studied groups. These inferences were based on the relationships between the variables measured in the questionnaire and the subsequent use of statistical techniques. A cross-sectional study design was proposed, which focused on describing the situation and contrasting hypotheses, considering the participation of teachers from both educational contexts.

Therefore, the following research problem was posed in this study: what is the level of competence of university teachers in both contexts in relation to the use of digital technologies, based on the DigCompEdu Framework?

### 2.1. Objectives of the Research

The purpose of this study was to determine the level of digital competence of teachers at two universities (Seville (Spain) and Arequipa (Peru)) (O1) and to identify whether there were statistically significant differences with respect to the level of self-perceived teaching digital competence (SDC) (O2).

### 2.2. Sample

The study sample was composed of a total of 2466 university teachers, including 1200 men (48.7%) and 676 women (27.4%), belonging to a Spanish university (Seville) (808) and a Peruvian university (Arequipa) (1658), most of them between 30 and 39 years old (22.9%) or between 40 and 49 years old (23.4%). A stratified random probability sample was used, which was stratified according to sex, age, and level. In addition, a high confidence level (90/95%) and a minimum sampling error (±2/3%) were considered. The age and years of experience data are shown in Table 1.

**Table 1.** Professional experience and experience implementing ICT.

| | | UNIVERSITY | |
|---|---|---|---|
| | | **Seville** | **Arequipa** |
| | | **Count** | **Count** |
| | 1–3 years | 8 | 377 |
| | 4–5 years | 38 | 248 |
| | 6–9 years | 86 | 310 |
| Professional experience | 10–14 years | 252 | 309 |
| | 15–19 years | 288 | 184 |
| | 20 or more years | 136 | 230 |
| | I do not use technology as an educational tool. | 62 | **547** |
| | | 30 | 464 |
| | Less than 1 year | 64 | 284 |
| Experience implementing ITC | 1–3 years | 102 | 225 |
| | 4–5 years | 92 | 66 |
| | 6–9 years | **458** | 0 |
| | 20 years or more | 0 | 72 |

Regarding how much experience the teachers in Seville had using ICT, most of them have been using it for 20 years or more, while in Peru, the largest group of teachers indicated that they do not use technology as an educational tool, and this was consistent with their professional experience, as most of them claimed to have 1–3 years of experience. The results seem to indicate that teachers belonging to the University of Peru are lacking when it comes to using and applying ICT in their educational practice.

### 2.3. Data Collection Instrument

For data collection and the subsequent analysis, the "DigCompEdu Check-In" questionnaire was used. It is considered the instrument of reference, as it has been corroborated by several expert authors in the field [45]. This instrument, validated by [43], was selected

as the primary instrument for assessing the level of university DTC and was validated by means of structural equations by [40].

Concerning the reliability of the instrument, they obtained Cronbach's alpha values for the set of items (0.937) (Table 2) and its component dimensions (professional commitment (0.813), digital resources (0.755), digital pedagogy (0.978), assessment and feedback (0.863), student empowerment (0.925), and the facilitation of students' digital competence (0.914)) (Table 3), corroborating that the instrument is sufficiently robust to discriminate the level of digital competence of the subjects [40]. The validation described the existing reality, explaining the educators' perception of the importance of the analyzed subject as well as the importance of its applicability for their future professional development.

**Table 2.** Cronbach's alpha.

| Reliability Statistics | |
|---|---|
| **Cronbach's Alpha** | **No. of Elements** |
| 0.950 | 22 |

**Table 3.** Reliability dimensions.

| Dimensions of Reliability | | | |
|---|---|---|---|
| | **Mean** | **Dev.** | **N** |
| A | 2.1578 | 0.81384 | 808 |
| B | 2.3333 | 0.75536 | 808 |
| C | 2.0619 | 0.97881 | 808 |
| D | 1.7814 | 0.86309 | 808 |
| E | 1.8639 | 0.92500 | 808 |
| F | 1.8594 | 0.91461 | 808 |

The purpose of the instrument is to improve a teacher's understanding of the framework, providing them with an assessment of their skills, which is essential to achieve a highly competent mastery in their educational practice.

Teachers should fill in the self-assessment questionnaire using a Likert-type scale (1—not at all, 2—very little, 3—a little, 4—some, 5—quite a lot, and 6—a lot). A total of 22 items are distributed in six competence areas examined by DigCompEdu, which are related to the different competence areas that compose it: professional engagement (4 items), digital resources (3 items), teaching and learning (4 items), assessment and feedback (3 items), empowering learners (3 items), and facilitating the learner's DC (5 items). Once the questionnaire is completed, the tool generates a detailed and personalized report on the educator's level of competence in different domain areas.

## 3. Results

For a correct interpretation of the results, it should be considered that the response interval ranges between 0 and 4. As for the mean scores, Table 4 shows the mean values and standard deviations achieved by the participating teachers for the different questions by dimension and globally.

In terms of digital competence in Seville, the values range from basic (1.36) to intermediate (2.30). Teachers show problems (at the basic level) in empowering their learners to act safely and responsibly on the Internet; employing digital assessment tactics to monitor students' progress; and making use of digital technology tools to provide effective feedback. The competences that stand out (at the intermediate level) are I develop my own digital materials and make changes to existing ones in order to adapt them to the demands and requirements I have as a teacher; I systematically use various digital media in order to improve communication with my colleagues and students; and I teach my students to collaborate in groups or teams, using digital technology tools to obtain and record information and knowledge.

**Table 4.** Means and standard deviations of items, dimensions, and digital competence.

| | UNIVERSITY | | | |
| --- | --- | --- | --- | --- |
| | Seville | | Arequipa | |
| | **M** | **SD** | **M** | **SD** |
| **A1**. I systematically use various digital media to improve communication with my colleagues and students. | 2.31 | 0.818 | 2.63 | 0.753 |
| **A2**. I make use of digital technology tools to collaborate with colleagues both inside and outside the educational institution where I work. | 2.18 | 0.947 | 2.33 | 960 |
| **A3.** I make use of digital technology tools to collaborate with colleagues both inside and outside the educational institution where I work. | 2.16 | 1.137 | 2.29 | 0.871 |
| **A4**. I make use of digital technology tools to collaborate with my colleagues both inside and outside the educational institution where I work. | 1.99 | 1.250 | 2.82 | 0.957 |
| **B1**. I make use of digital technology tools to collaborate with my colleagues both inside and outside the educational institution where I work. | 2.21 | 0.962 | 2.40 | 0.807 |
| **B2**. I develop my own digital materials and make changes to existing ones in order to adapt them to the demands and requirements I have as a teacher. | 2.49 | 0.858 | 2.59 | 0.736 |
| **B3**. I develop my own digital materials and make changes to existing ones to adapt them to the demands and requirements I have as a teacher. | 2.30 | 1.130 | 2.53 | 1.040 |
| **C1**. I carefully analyze when, how, and why to use digital technology tools in the classroom to ensure that I obtain the maximum benefit from their added value. | 2.12 | 1.190 | 2.61 | 0.961 |
| **C2**. I supervise the interactions and tasks my students carry out online and in the collaborative spaces we use to ensure a safe and rewarding environment. | 1.99 | 1.207 | 3.09 | 0.779 |
| **C3**. When students collaborate in groups or teams, they use digital technology tools to obtain and record information and knowledge. | 2.22 | 1.302 | 2.84 | 0.918 |
| **C4**. I make use of digital technology tools to give students the ability to plan, document, and evaluate their own learning process. Examples of this include self-assessment tests, digital portfolios, blogs, and forums. | 1.91 | 1.130 | 2.67 | 0.822 |
| **D1**. I use assessment tactics in the digital environment to monitor students' progress. | 1.76 | 0.997 | 2.59 | 0.800 |
| **D2**. I use assessment tactics in the digital environment to monitor students' progress. | 1.73 | 1.044 | 2.56 | 0.968 |
| **D3**. I make use of digital technology tools to provide effective feedback. | 1.85 | 1.002 | 2.58 | 0.896 |
| **E1**. When designing digital activities, I consider and address potential problems that may arise, such as inequity in access to digital devices and resources, compatibility issues, and a low level of digital competence on the part of learners. | 1.97 | 1.384 | 3.01 | 0.980 |
| **E2**. I use digital technology tools to provide students with personalized learning opportunities, such as assigning different digital tasks to meet their individual learning needs and taking into account their preferences and interests. | 1.42 | 1.382 | 2.37 | 1.222 |
| **E3**. I make use of digital technology tools to encourage the active participation of students in the classroom. | 2.20 | 1.051 | 2.61 | 0.933 |

**Table 4.** *Cont.*

| | UNIVERSITY | | | |
| --- | --- | --- | --- | --- |
| | Seville | | Arequipa | |
| | **M** | **SD** | **M** | **SD** |
| **F1**. I teach students how to assess the reliability of information searched online and how to identify erroneous and/or biased information. | 1.93 | 1.113 | **2.28** | 0.927 |
| **F2**. I propose tasks that require students to use digital media to communicate and collaborate with each other or with an external public body. | 1.91 | 1.076 | 2.56 | 0.857 |
| **F3**. I propose activities that require students to produce digital content, such as videos, audio, photographs, presentations, blogs, and wikis, among others. | 2.14 | 1.204 | 2.64 | 0.941 |
| **F4**. I teach students how to act safely and responsibly on the Internet. | **1.36** | 1.145 | 2.43 | 1.004 |
| **F5**. I motivate students to creatively use digital technology tools to address specific problems, such as overcoming obstacles or responding to challenges in their learning process. | 1.96 | 1.083 | 2.62 | 0.899 |
| **Area A**—Professional Engagement | 2.29 | 0.850 | 2.64 | 0.696 |
| **Area B**—Digital resources | 2.32 | 0.799 | 2.50 | 0.708 |
| **Area C**—Teaching and Learning | 2.21 | 1.002 | 2.93 | 0.746 |
| **Area D**—Assessment | 1.78 | 0.888 | 2.58 | 0.795 |
| **Area E**—Empowering Learners | 1.87 | 1.145 | 2.67 | 0.895 |
| **Area F**—Facilitating Learner's Digital Competence | 1.84 | 0.967 | 2.51 | 0.787 |
| **Digital Competence Total** | **2.01** | **0.837** | **2.59** | **0.662** |

In the case of Peru, the values are between the intermediate level (2.28) and the medium level (3.09), somewhat higher than the maximum level in Seville. Specifically, the least developed competences are I teach students how to assess the reliability of information searched for online and how to identify erroneous and/or biased information and I make use of digital technology tools to collaborate with my colleagues both inside and outside the educational institution where I work. On the other hand, the competences that stand out are I monitor the interactions and tasks my students carry out online and in the collaborative spaces we use as a means of ensuring a safe and rewarding environment and when I design digital activities, I consider and address potential problems that may arise, such as inequity in access to digital devices and resources, compatibility issues, and a low level of digital competence on the part of students.

It is very significant that, in both countries, the competences that stand out are in the first two areas, while the least developed competences correspond to areas in which the evaluation of the teaching–learning process (area 4), attention to diversity (area 5), and the development of digital citizenship (area 6) predominate.

Specifically, the mean value achieved for the instrument overall for Seville is 2.01 points, with a deviation of 0.84. This indicates that, in general, the level of competence is intermediate. In the case of Peru, the mean value achieved for the instrument overall is 2.59 points, with a deviation of 0.67, a result very similar to that obtained in Seville. We could say that teachers have basic levels of DTC, although they consider themselves more competent in professional engagement and in the development of digital resources, both in Seville and in Peru. This last assumption is very significant.

For the comparative analysis, participants were asked to rank their self-perceived level of digital competence (Table 3), both at the beginning (pre) and at the end (post) of the questionnaire. The ranking system was as follows: newcomer (A1), explorer (A2), integrator (B1), expert (B2), leader (C1), and pioneer (C2).

For the analysis of the obtained data, the Mann–Whitney U-test (Table 5) and the Wilcoxon test, a non-parametric statistical test, were applied to check if there were differ-

ences according to the level of competence of the university teachers, that is, to find out whether there were significant differences between the measurements taken before and after the intervention.

**Table 5.** Self-perception of DTC level.

|  | PRE_CD | DIM_A | DIM_B | DIM_C | DIM_D | DIM_E | DIM_F | P_TOTAL | POST_CD |
|---|---|---|---|---|---|---|---|---|---|
| **Mann–Whitney-U** | 635,679.000 | 506,856.000 | 581,888.000 | 396,194.000 | 343,784.000 | 399,715.000 | 404,905.000 | 402,325.000 | 653,208.000 |
| **Wilcoxon** | 962,515.000 | 833,692.000 | 908,724.000 | 723,030.000 | 670,620.000 | 726,551.000 | 731,741.000 | 729,161.000 | 2,028,519.000 |
| **Z** | −2.158 | −10.624 | −5.791 | −17.453 | −20.768 | −16.901 | −16.915 | −17.408 | −1.052 |
| **Asymptotic Sig. (bilateral)** | 0.031 | 0.000 | 0.000 | 0.000 | 0.000 | 0.000 | 0.000 | 0.000 | 0.293 |

a. Grouping variable: University.

The results obtained in the pre-test show that there were significant differences at both universities in relation to the global conception of the term (0.031). We observed that they had very high conceptions of professional teaching commitment, in the creation of digital content and resources, and concerning the capacity to develop actions that generate an increase in students' digital competence.

## 4. Discussion

Thanks to its large sample size, this study has provided a luminous vision of different phenomena, giving an overview of the university teaching profile in both contexts.

Firstly, looking at the career paths of the participants, we can affirm that the differences between the two universities are striking, with the teachers from Seville being those who have been using ICT as a methodological tool for the longest time (20 years or more) since, as detailed above, Spain is one of the countries with the greatest teaching preparation in this field, which is not the case in the Peruvian context, where they claim to have no more than 3 years of experience in the use of educational technologies. According to the studies by [20], this is due to the need to strengthen actions using educational policies in order to establish a strategic plan through digital training programs focused on the development of the different competency dimensions. Moreover, this situation also generates inequalities among teachers due to the lack of skills and resistance to incorporating new learning scenarios, which is confirmed in studies where the variable "professional experience" reflects a decrease in teachers' competence levels [49].

These basic results may be due to the importance given by institutions. In other words, despite the years implementing ICT, Spain is one of the countries where the use of ICT in teaching is least promoted (53.3% compared to the OECD average of 67.2%) [17]. The same is true for Peru, where the focus is more on student competences, leaving aside teacher training. Therefore, the adaptation of frameworks to the needs of specific contexts is recognized as an existential requirement. This would include, as [19] mention in their research, a concretization, for example, of digital resources for evaluating educational practice.

If we analyze the most outstanding competences, we can observe that in Seville, teachers promote the exchange of learning experiences, which leads to the creation of new ways of teaching and, therefore, to educational innovation; the use of various digital channels for educational innovation; improving communication inside and outside the classroom; and the encouragement of collaborative learning through the use of tools that allow them to monitor students' progress and advancement in greater detail, adapting to the situation and context of each classroom. These findings suggest, as mentioned by [50,51], that teachers are evolving from a digital perspective focused on the use of technological resources towards broader conceptions. These conceptions include knowledge generation through the use of advanced communication technologies (ACTs) and participation in collaborative environments through educational and pedagogical technologies (PETs). In contrast, in Peru the standing competences are different. The focus is more on technical rather than pedagogical aspects, emphasizing the importance of supervising spaces to ensure student safety as well as the development of skills to solve problems related to

accessing and using digital devices. Regarding the least valued competences, which are located at a more basic level, teachers show more problems when it comes to enabling their students to act safely and responsibly on the Internet; to employ digital assessment tactics to monitor students' progress; and to make use of digital technology tools to provide effective feedback. In this sense, it is clear that adequate teacher training is required, as a lack of competency mastery implies an education that is not geared towards the most sought-after professions in the 21st century [19].

Another aspect to detail is the self-perception, as this was higher in both contexts before the questionnaire was carried out than after reflection about digital competence.

This may be due, as different studies have shown, to the fact that teachers often have an idealized view of their own digital skills, which influences their self-perception of competence [52,53]. However, by completing and reflecting on a questionnaire that assesses their digital competence, they may experience a change in the way they see themselves. This reflection may lead them to have a more realistic perception of their skills and knowledge in the digital domain, as it can be observed, once the test has been taken again, that there are no statistically significant differences in the post-test, with no higher score. This is due to the fact that the teacher does not rate his or her ability to use ICT highly after understanding and reflecting on his or her self-perception. In the case of Spanish teachers, this is because the more training they have, the higher their self-perception [22], and in the case of Peru, it is due to the ineffectiveness of teacher training, the bureaucracy existing in the organizations, and the lack of support from specialized staff for the implementation of the DDC [20], which means that teachers do not make critical judgements of their abilities until they are able to reflect on their educational practice due to the lack of digital training [16].

The obtained findings allow us to understand the level of digital competence of university teachers regarding the use of digital technology and innovation in different educational environments based on the Common Digital Competence Framework for Teachers. This has been possible thanks to the reliable data collection instrument used to measure teachers' digital competence, including the level of knowledge, the use of digital resources, and participation in online teacher training and innovation projects.

This research opens doors for future studies, offering a more complete and representative vision by taking into account different educational contexts. As future lines of research, it is proposed, as a result of the data offered, to identify the variables that can predict the level of digital competence of teachers as well as to design specific training actions aimed at improving teachers' digital competence, as detailed in the studies by [54] and in the teaching practices carried out by [55].

## 5. Conclusions

According to what this work has shown and the research carried out by several authors, such as [56–59], it has been demonstrated that the digitalization of society has had a significant impact on the educational sphere. This phenomenon has broken down the barriers of knowledge, allowing its access in a flexible and broad way. However, it is not enough to have an abundance of information; it is crucial to understand how, where, why, and for what purpose it will be used. It is essential to promote teachers' digital literacy to achieve quality, equity, and excellence in education [60,61].

As evidenced throughout this research, the expansion of technology in education requires that educational actors acquire significant digital competencies to address evolving educational needs [62]. This implies the need to keep up to date and adapt to the changing demands of the educational environment, which increasingly requires the effective use of digital tools and the ability to make the most of the opportunities they offer.

To meet these demands and provide quality education that is adapted to current needs, common reference frameworks have been proposed. These frameworks aim to support holistic learning that lays the foundations for the creation of innovative and meaningful educational practices and to promote digital developments that are in tune with today's and

tomorrow's societies. In this way, they aim to ensure that education systems are prepared to face the challenges and opportunities posed by the ever-changing digital environment.

This study on digital competence in teaching serves as a starting point for future research in this area, providing an understanding of the level of digital competence of university teachers in a variety of contexts. These findings are in line with another study focusing on this topic [63–65]. The analyses and results that were obtained underline the importance of carrying out context-specific, customized training to improve teachers' digital competence. Specific training plans need to be strengthened and adapted to each educational stage to achieve these objectives [66]. It is essential that regional, national, and European policies are in tune with each other to incorporate knowledge and experience in the digital field and to facilitate convergence in the creation of a European Education Area in 2025 that promotes the appropriate use of ICT.

In terms of limitations, it is important to consider the following aspects for future research. Firstly, it would be advisable to include probabilistic samples for each area to be investigated, which would allow more representative and extrapolated data to be obtained at a global level. It is also recommended to broaden the geographical scope of the study to include different educational contexts and not be limited to specific institutions. Another aspect to point out is the importance of considering uncontrolled variables that may influence the results, such as the individual characteristics of teachers, the technological resources available, and the socio-economic factors of the different environments. For future work, it is suggested to extend the study to other types of educational institutions, such as non-university-level institutions, in order to obtain a more complete and representative picture, emphasizing the following aspects:

- Analyzing the significant differences between the studied groups in terms of teachers' digital competence.
- Identifying the variables that can predict the level of teachers' digital competence.

Finally, another very useful aspect would be to obtain real or informal testimonials from recognized educational institutions in the field that could support and validate the progress achieved using ICT in education. These testimonials and reports would provide specific evidence of the benefits and positive impacts that ICT may have on education.

To summarize, the rapid evolution of technologies has a significant impact on the work of teachers, which can lead to difficulties in updating and applying teaching to current learning needs. This change highlights the need to adapt teacher training to real demands, which require an approach beyond traditional master teaching. Therefore, it is essential to acquire competences on the correct use of ICT to implement it effectively into teaching and learning processes. It is necessary to adjust teacher training to meet current demands and provide quality teaching that promotes meaningful student learning.

**Author Contributions:** L.M.-P., C.L.-C. and J.B.-O. presented and designed the experiments; L.M.-P. performed the experiments, analyzed the data, and wrote the original article. C.L.-C. and J.B.-O. contributed to the review and editing. All authors have read and agreed to the published version of the manuscript.

**Funding:** This research did not receive any external funding.

**Institutional Review Board Statement:** The ethical review and approval for this study was waived because the subjects participating in the study (1659) responded to the signed consent form before answering the questionnaire.

**Informed Consent Statement:** Informed consent was obtained from all subjects involved in the study.

**Data Availability Statement:** Due to confidentiality and privacy agreements, it is not possible to make these data publicly available.

**Conflicts of Interest:** The authors declare no conflict of interest.

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
