# Peer review of "Self-Perception of Digital Competence in University Lecturers: A Comparative Study between Universities in Spain and Peru According to the DigCompEdu Model"

_societies, doi:10.3390/soc13060142_

Round 1
Reviewer 1 Report
Dear authors,
My comments are:
In the abstract, please describe the research gaps and the implications of your study. In the abstract, avoid using METHOD, SAMPLE, RESULTS, and DISCUSSION.
Why would you compare Peru and Spain?
2. Materials and methods: Please recheck again the whole writing.
2.3 Data collection: It is completely impossible for a construct to have a Cronbach's alpha greater than 1 (which student empowerment have value of 1.08).
Table 3: Please revise the Mann-Whitney and Wilcoxon tests in the table's left column.
Discussion is superficial and uncritical.
Overall, it is unclear why this research is necessary. Also unclear is its potential contribution to the body of knowledge. The methodology and pilot research are slack. The discourse is superficial.
Thank you. Hopefully the author will find the feedback helpful.
Author Response
Thank you for your feedback on the article. All the detailed improvements have been taken into account. You will find changes in the summary, in the importance of comparing the different contexts, in the analysis of the data obtained and in the argumentation of the proposal. Regarding the data collection, the value is 0.925 instead of 1.08. This is due to a transcription error. We hope that it meets the demanded requirements.
Reviewer 2 Report
The paper is coherent and its main topic is interesting and relevant for the scientific community. At first glance, it seems to offer a significant perspective on the study of teacher digital competence, with a great contribution to the educational community. However, it has a considerably improved research design to meet the scientific rigor required by the journal.
It is recommended:
Review the IMRyD structure within the summary.
Justify the use of non-parametric contrast statistics and clarify in the abstract that the study is descriptive-inferential.
Provide reliability and validity data of the instrument.
Review citation and referencing in depth.
Even if the quality of the English language is good enough to be published, it would be strongly recommened a wording review, especifically regarding orthotypography issues.
Author Response
Thank you for your feedback on the article. All detailed improvements have been taken into account. You will find changes in the abstract, the justification, the data for reliability and in the citations and references of the article. Regarding the language improvement, it has been reviewed by native speakers. We hope that it meets the requirements demanded.
Reviewer 3 Report
The article “Self-perception of university lecturers in digital competence: a comparative study between universities in Spain and Peru according to the DigCompEdu model” is coherent.
The topic is very interesting and covers a relevant topic. At first glance, it seems to offer a significant perspective on the study of teacher digital competence, with a great contribution to the educational community.
The document is well-structured, easing the comprehension of the study conducted.
The literature review is relevant. However, it is recommended to add new references that highlight the research to date on the subject (teaching digital competence).
The research problem and the objectives of the study are well defined and clearly achievable.
The research phases are presented in a clear and structured way. The results are presented in a descriptive way for each dimension of the instrument, which facilitates understanding.
The discussion and conclusions are clearly specified and respond to the objectives of the study.
Author Response
Thank you for your feedback on the article. All recommended improvements have been taken into account. You will find changes in: summary and literature, where new references have been added. We hope that it meets the requirements.
Reviewer 4 Report
The manuscript entitled “Self-perception of university lecturers in digital competence: a 2 comparative study between universities in Spain and Peru according to the DigCompEdu model” employed a descriptive comparative research model in order to investigate perceived digital competence of university instructors in a context of higher education institutions. Framed by the DigCompEdu, the stduy addresses an important and interesting topic, which aligns with the scope and objectives of the journal. The study sample consists of 2466 university teachers from Spain and Peru. The results illustrated implications highlight some institutional and training policy domains to be addressed regarding Digital Competences.
My remarks can be found in the PDF file as annotated notes and the bulleted notes in the regarding the list below.
1. Abstract
The abstract does not meet journal standards containing inappropriate terminology and grammar mistakes. The results and discussion part are missing and confusing.
2. Introduction
- In the introduction part the problem, the need for universities to develop the digital competence of their teaching staff, is clearly stated and helps the reader understand the importance of the issue and the area of education in which the problem lies.
- The introduction does not provide a clear rationale for why the study was conducted in two university from different countries. This is already done partly, yet there is a need for further elaboration.
- In this part, there is a need for further synthesis of previous studies and clarification on what makes the present study different from the previous studies and building on the previous research what contributions and originality it adds to the literature.
- The introduction section could be clearer and more concise in conveying its main ideas. There are some sentences that are complex, which may make it difficult for readers to follow the argument.
- Additionally, the introduction could benefit from a stronger concluding paragraph that summarizes the main points and highlighting the potential contributions of the paper to the literature.
There are some grammatical errors and awkward phrasing that could be revised for clarity.
- Instead of the methods section, the research questions or the hypotheses of the study could be stated in here.
3. Material and Methods
In the methods part, the authors mentioned that they employed a descriptive comparative design as a methodological framework. The methodological approach and research design employed in this study are widely accepted and appropriate for descriptive research studies. The authors also employed a stratified sampling technique, which increases the robustness of the results. However, more details about the data analysis procedures should have been presented.
4. Results
The findings should have been more effectively communicated in a clear and comprehensible manner. There is too much data, but little results are presented. Nevertheless, while there is a little paucity of the presentation of the descriptive results, the results were insightful.
Additionally, the authors employed non-parametric tests. The rationale for these tests should be explained in the methods section, which is missing. Thus, it makes difficult for reviewers to come to conclusion regarding the appropriateness and robustness of the findings as well as the interpretation of the results.
The synthesis of results with previous studies was also proved to be satisfactory.
6. Discussion and Conclusion
-This part should start with an overall summary of the aim and the results of the study, which is missing. Then, the authors should present their main conclusions extracted from the research results.
- In the conclusion section, the conclusions and recommendations drawn based on the findings can be presented in bulleted format to grasp more attention. What are the main conclusions or learnt experiences extracted from the results by contrasting two universities that can be presented as implications for policy makers, university teachers, administrators, or students in higher education context?
- In addition, the limitations of the present study and recommendations for future research should be elaborated more and given in a more concrete way to set direction for future research.

There is a critical problem with the manuscript, which is the use of English through the text is not satisfactory for such a research article. There are many grammatical mistakes including incomplete sentences (e.g. line 189). Thus, I recommend a proofread by native speakers.
Author Response
Thank you for your feedback on the article. All recommended improvements have been taken into account. You will find changes in the summary, introduction, where the choice of the study sample is clarified, research questions, etc... On the other hand, there are also changes in the explanation of results, discussion and conclusions, as well as new references for a better understanding. With regard to the improvement of the language, it has been revised by native speakers. We hope that it meets the requirements demanded.
Round 2
Reviewer 1 Report
Why compare Peru and Spain
"This work aims to explore the educational reality and understand the importance of measuring and assessing the level of DC of university teachers in different contexts, specifically in Spain and Peru." - At least three citations are required to substantiate this claim. Without citations, the manuscript is of questionable quality.
Materials and methods: Please recheck again the whole writing.
The writing on this subtopic remains rather superficial. To ensure that a subtopic's quality is sufficient for publication in a journal that is indexed, authors must elaborate on it further. Your research methodology is likewise vague.
Table 3: Please revise the Mann-Whitney and Wilcoxon tests in the table's left column.
"W de Wilcoxon" is still lacking something that seems appropriate.
Discussion is superficial and uncritical.
The discussion is largely unchanged, and the authors have not made many changes. The discussion does not merit publication in a journal that is indexed. Extensive amendments are required.
I advise the authors to include two more subtopics, namely Practical Implications and Theoretical Implications, to the discussion.
Thank you.
Author Response
All requested modifications have been taken into account. The document is included. Please see the attachment.

Reviewer 4 Report
Dear Authors,
Congratulations for your hard work and improved manuscript. Yet there are some minor details needs further revisions as listed below.
- line 43: CD or DC?
- line 94: What does CDD stand for?
- line 208: I advise to change the title from Test sample to Sample?
- line 223: The final line of the table 1, there is a "7" in a separete cell? What is it?
- line 223: In the table 1, Is it "Experience implementing ITC" or ICT?
- line 305: In the table 5, Is it "W de Wilcoxon" or Wilcoxon-W?
- lines 101 and line 236, : The verb "analyze" is written in US English but in lines 14, 333, and 391, it is written in UK English. It should be consistent through the text.
- Through the text, the word 'education' is not a proper noun, so it shouldn't be written like Education (e.g. 376) .
Author Response
Rev.2
All requested modifications have been taken into account.
- Line 43: DC. Transcription error.
- Line 94: in Spanish means Competencia Digital Docente. There is a mistake in the transcription. Correct: DTC- Digital Teaching Competence.
- Line 208: changed to "Sample".
- Line 223: the 7 has been deleted.
- Line 223: the correct title is "Experience in the use of ICT".
- Line 305: W de Wilcoxon.
- Lines 101 and 236: the text is written in British English. Amended to read "analyse".
- Line 376: the word education has been modified throughout the text.
We hope that it complies with the requested improvements.
Round 3
Reviewer 1 Report
Thank you for the amendments.